# The Role of “Metabolic Instability” as a New Risk Criterion in the Pathogenesis of Endometrial Cancer

**DOI:** 10.3390/cancers17233840

**Published:** 2025-11-29

**Authors:** Maria A. Sukhanova, Sergey Yu. Muraviev, Evgeniy A. Tarabrin, Alexey L. Shestakov, Zelimkhan G. M. Berikkhanov, Irina A. Tarasova, Vadim S. Razumovsky, Ivan A. Markov, Andrey P. Fabrika

**Affiliations:** Sechenov First Moscow State Medical University, Department of Hospital Surgery 2, 119048 Moscow, Russia; muravev_s_yu@staff.sechenov.ru (S.Y.M.); tarabrin_e_a@staff.sechenov.ru (E.A.T.); shestakov_a_l@staff.sechenov.ru (A.L.S.); berikkhanov_z_g@staff.sechenov.ru (Z.G.M.B.); tarasova_i_a@staff.sechenov.ru (I.A.T.); razumovskiy_v_s@staff.sechenov.ru (V.S.R.); markov_i_a@staff.sechenov.ru (I.A.M.); fabrika_a_p@staff.sechenov.ru (A.P.F.)

**Keywords:** endometrial cancer, metabolic syndrome, metabolic instability, hormonal-metabolic lability, adipokines, adipose tissue dysfunction

## Abstract

New data suggest that endometrial cancer, traditionally associated with obesity and high estrogen levels, may not fully explain its occurrence in women with a normal body weight. Modern studies emphasize the importance of metabolic instability, including fluctuations in body weight, blood sugar levels, insulin production, adipokine levels, and sex hormone levels, in creating an environment conducive to carcinogenesis. These dynamic changes can lead to chronic inflammation, oxidative stress, and impaired cell repair, even in the absence of clinical signs of a metabolic disorder. This review summarizes the evidence that the repetitive variability of metabolic parameters, rather than single values, can serve as a sensitive indicator of the risk of endometrial cancer. The focus on temporal patterns rather than static measurements highlights the importance of monitoring a woman’s metabolic dynamics. A better understanding of these dynamics may improve early risk stratification for precancerous changes in non-obese women.

## 1. Introduction

Endometrial cancer (EC) is one of the most common malignant neoplasms of the female reproductive system, and its incidence continues to increase worldwide [1,2]. The risk factors for endometrial cancer are well established and multifactorial. They include age, hormonal and reproductive characteristics (early menarche, late menopause, anovulation, nulliparity), hereditary predisposition, including Lynch syndrome as well as exogenous exposure to estrogens without progesterone opposition. [2,3]. However, in recent decades, researchers have increasingly focused on the relationship between EC and metabolic syndrome (MS), which encompasses insulin resistance, hyperglycemia, arterial hypertension, and abdominal obesity. Together, these components form the so-called “endometrial cancer triad” (as described by Weiqin Zheng et al., 2024), creating an unfavorable metabolic background that not only promotes carcinogenesis but also affects the course and prognosis of the disease [2,4,5].

Traditionally, EC has been considered a “disease of obesity”—most patients are overweight or obese, and hyperestrogenism resulting from peripheral aromatization of androgens in adipose tissue is regarded as the key pathogenic mechanism [6]. However, growing evidence challenges this simple causal relationship between obesity and tumor growth. Increasingly, studies indicate that not only the quantity but also the metabolic activity, inflammatory state, and hormonal function of adipose tissue determine the risk of EC development.

Leonardo de Andrade Mesquita et al. (2023) highlighted a group of women who are overweight (BMI 25–29.9) but not obese, yet already exhibit signs of metabolic dysfunction—insulin resistance, dyslipidemia, and chronic subclinical inflammation [7]. This population, which constitutes a large proportion of the general female demographic, remains largely overlooked in oncogynecological research. Meanwhile, these women may demonstrate early markers of metabolic disturbances reflected in both biochemical parameters and phenotypic features, such as altered fat distribution or the presence of cellulite.

Despite extensive evidence supporting the role of obesity and diabetes in increasing the risk of endometrial cancer, several key aspects remain insufficiently understood. In particular, emerging data suggest that metabolic dysfunction, even in individuals with normal or borderline BMI, may serve as an independent risk factor capable of initiating carcinogenic processes in the endometrium. Furthermore, external and visual phenotypic manifestations reflecting the state of adipose tissue and its inflammatory remodeling, such as pronounced cellulite, local thickening of subcutaneous fat, and fluctuations in body weight, are attracting increasing attention. These features may represent potential clinical markers of metabolically associated oncological risk, reflecting internal alterations in adipose tissue even in the absence of overt obesity.

Thus, current concepts regarding the relationship between metabolic syndrome and endometrial cancer require reconsideration. Increasing emphasis is being placed not only on the quantitative but also on the qualitative characteristics of adipose tissue its metabolic activity, endocrine function, and local inflammation. Understanding these aspects opens new perspectives for risk stratification, early diagnosis, and disease prevention.

## 2. Materials and Methods

### 2.1. Study Design

The concept of “metabolic instability” has not been previously used in the scientific literature as an independent clinical or pathogenetic term. As a result, there are no studies aimed at assessing this phenomenon as an integral risk factor. Despite this lack of an initial terminological basis, the screening of the literature on endometrial cancer risk factors in women without obesity was conducted in accordance with the PRISMA guidelines [8]. This approach remained methodologically sound, as it ensured transparency in the selection process, the sequence of steps, and the criteria for including and excluding sources.

The review presented in this paper has a comprehensive nature, meeting the key standards of systematic analysis, such as structured search, multi-stage selection, quality assessment, and reproducibility of methodology. However, it combines data on several interrelated groups of risk factors, including metabolic, hormonal, and inflammatory, rather than focusing solely on one specific definition.

As a result, this work represents a systematic review of scientific literature on endometrial cancer risk factors among women without obesity. The synthesis stage of the study identified metabolic instability as a potential mechanism underlying carcinogenesis.

The methodological basis of this study was further aligned with the principles of AGREE II (Appraisal of Guidelines for Research & Evaluation II) and the evidence criteria of the Oxford Centre for Evidence-Based Medicine (2011), in order to increase the transparency, reproducibility, and validity of our choice of evidence. A block diagram of the research selection process is shown in Figure 1.

### 2.2. Search Strategy

A comprehensive literature search was conducted in *PubMed*, *Embase*, and *Google Scholar* from inception to October 2025. Search terms included combinations of the following keywords and MeSH terms: “endometrial cancer”, “metabolic syndrome”, “metabolic instability”, “adipokines”, “obesity”, “insulin resistance”, “leptin”, “adiponectin”, “estrogens”, “sex hormone-binding globulin”, “hormonal fluctuations”, and “hyperandrogenism”. Boolean operators (AND, OR, NOT) were applied to optimize the search strategy. All found publications were imported into the Rayyan QCRI platform, which made it possible to automatically eliminate duplicates and standardize the selection procedure.

### 2.3. Inclusion and Exclusion Criteria

The final analysis included publications that met pre-defined criteria in terms of subject matter, design, and level of evidence. The main criterion for inclusion was the focus on assessing the relationship between metabolic, hormonal, and inflammatory factors and the risk of endometrial cancer.

We selected papers that analyzed women without obesity (body mass index [BMI] < 30 kg/m^2^) or provided stratified data, allowing us to identify a group with normal or overweight body weight. Therefore, the review included studies that could assess the contribution of metabolic instability, hormonal imbalance, and adipose tissue dysfunction, outside the influence of morbid obesity.

According to the type of study design, the analysis included original clinical studies, cohort, cross-sectional, and case–control studies, providing the opportunity for quantitative or qualitative assessment of the relationship between studied factors and oncological outcomes. Additionally, secondary studies were considered if they met certain criteria, such as sufficient methodological transparency and validity of conclusions. These included systematic reviews, meta-analyses, and qualitative reviews. All publications included in the analysis had to be peer-reviewed, containing a description of study design, inclusion and exclusion criteria, sample characteristics, and statistical methods. This approach helped minimize methodological differences between studies and allowed for comparison of results by key parameters.

Publications that did not meet the required levels of evidence or completeness of methodological description were excluded from the analysis. This included single clinical observations (case reports), small case series without control groups, expert opinions, and works that lacked sufficient data for further extraction and analysis. Additionally, studies involving only obese women (BMI ≥ 30 kg/m^2^) were excluded, as were those that did not allow for identification of subgroups with lower body weight. Studies that did not focus on metabolic, endocrine, or inflammatory mechanisms of carcinogenesis or molecular genetic studies unrelated to systemic metabolic disorders were also excluded.

Works with insufficient methodological quality were also excluded from the analysis. These included materials that had not undergone peer review, such as student collections and unpublished local reports, as well as studies that failed to meet quality assessment standards set by the Oxford Center for Evidence-Based Medicine and AGREE II. Works that did not provide information on the study design, sample characteristics, statistical analysis methods, or criteria for participant selection were also excluded. Additionally, works for which the full data was unavailable, such as materials presented only as conference abstracts or presentations without full-text publications, were also excluded.

Finally, duplicate publications were excluded from the analysis. This included repeated reports of the same data as well as expanded or abbreviated versions of studies that did not provide additional significant information. By following a step-by-step selection process, we ensured consistency, transparency, and adherence to PRISMA, Oxford, and AGREE II guidelines. This approach also minimized the risk of methodological bias and selection bias.

### 2.4. Selection Process and Quality Assessment

The initial screening was conducted by two independent researchers, and the stages included reviewing titles and abstracts, as well as evaluating the full texts. Any disagreements were resolved through consensus with the involvement of a third expert.

In order to increase analytical rigor, we used several tools: Rayyan QCRI for blind selection and elimination of duplicate studies, AGREED II for standardizing evidence quality assessment, and Oxford Center for Evidence-Based Medicine’s (OCEBM) classification system for determining the level of evidence.

After applying these methods, we selected a total of 57 publications that met the criteria for methodological quality and relevance for our final analysis. To ensure transparency and reproducibility, we have presented all included studies in a tabular format.

Table 1 shows the original studies that were included, including the author(s), year of publication, study design, and sample size.

Table 2 Included reviews (author, year, type of review—systematic, meta-analysis, narrative; purpose of the study).

### 2.5. Data Extraction and Synthesis

Due to the lack of a unified definition and a consistent terminology for the concept of “metabolic instability”, the studies included in this review differ in their design, assessment methods, and the range of biological markers analyzed. Given these differences, it would be inappropriate to use classical quantitative methods of data synthesis, as it could potentially lead to biased results.

Instead, we have chosen to apply a structured narrative approach, which is in line with the PRISMA guidelines. This method allows us to organize the research findings into different themes, compare pathophysiological mechanisms presented in different studies, assess the consistency of data, and identify areas where scientific knowledge is lacking. By including clinical, metabolic, inflammatory, and hormonal parameters, we aim to provide a comprehensive analysis of the available evidence.

To increase transparency and ease of interpretation, the publications were grouped into four analytical categories: (1) Metabolic mechanisms and risk factors; (2) Endocrine changes and hormonal regulation; (3) Inflammation and adipokines; (4) Clinical features of endometrial cancer in women without obesity.

This synthesis format allows us to not only summarize heterogeneous data, but also, for the first time, consider the totality of the identified changes as a single potential pathogenic contour underlying the risk of endometrial cancer in women without obesity.

### 2.6. Definition of Key Terms

The following working definitions were used to maintain terminological consistency:−Metabolic instability is a dynamic condition characterized by frequent chaotic repetitive fluctuations in body weight, glucose, insulin, sex hormones and adipokines, in the absence of stability of homeostasis, leading to chronic cellular stress, impaired adaptive mechanisms and progressive subclinical inflammation in body tissues.−Hormonal and metabolic lability is a variable interaction of hyperinsulinemia, relative hyperandrogenism, changes in peripheral receptor sensitivity and adipokine imbalance, which forms an unstable endocrine environment of multidirectional (oppositely directed) stimulation of the endometrium, which increases the likelihood of proliferative disorders and replicative.

## 3. Results

The findings of this systematic review are summarized below and organized into thematic subsections addressing the epidemiological, metabolic, and hormonal aspects of endometrial carcinogenesis in non-obese women. Each subsection highlights the most consistent results identified across 57 studies included in the review, following the PRISMA 2020 framework. Across studies, metabolic instability was consistently associated with increased endometrial proliferative activity, chronic subclinical inflammation, and altered adipokine signaling. The direction of effect was uniform, indicating a potential pro-carcinogenic role of metabolic and hormonal variability even in the absence of obesity.

### 3.1. The Concept of Metabolic Instability as an Integral Characteristic of Risk

Based on our clinical experience and the analysis of included studies, we have found that the most significant changes in endocrine, metabolic, and inflammatory parameters in non-obese women do not occur during stable metabolic states, but rather during the transitions between them.

This has led us to identify a phenomenon we have termed “metabolic instability”, which can be described as a kind of “ arrhythmia” of physiological processes during body adaptation to fluctuations in body weight caused by irregular diets, changes in physical activity, and lifestyle. An example of this is when a woman attempts to lose weight repeatedly, breaks her diet, and then regains weight. Such chaotic and repetitive changes in physiological parameters that occur during the transition from weight loss to weight gain, and vice versa, without a stabilization phase, create a state of metabolic instability. Under these circumstances, the body is unable to achieve stable homeostatic balance, which itself is a risk factor for error, and the regularity of these changes creates a background for dysregulation and, consequently, pathological processes.

Unlike stable metabolic disorders, such as insulin resistance, chronic hyperinsulinemia, or persistent subclinical inflammation, metabolic instability is a process in which the body is constantly rebuilding the work of the energy, hormonal, and immune systems. These adaptive mechanisms, with rare and smooth changes, are accompanied by the gradual formation of a new homeostatic level. However, due to frequent fluctuations in nutrition, activity, or weight, which are typical of modern women’s lifestyles, these systems operate in a nearly continuous switching mode between anabolic and catabolic programs.

This transitivity is reflected in several physiological domains simultaneously:−Endocrine: Changes in the secretion of insulin, leptin, adiponectin, sex hormone-binding globulin (SHBG) levels, and the bioavailability of estrogen.−Immune system: Short-term peaks in pro-inflammatory cytokines such as IL-6, TNF-α, and MCP-1, which do not have time to stabilize.−Metabolic: Indicators of lipolysis and oxidative stress increase during weight loss phases, followed by increased lipogenesis and aromatase activity during weight gain phases.

Thus, metabolic instability is not a disorder of the endocrine system or inflammation in itself, but rather a complex, supra-systemic condition that occurs at the intersection of various physiological processes. An important aspect of this phenomenon is its behavioral nature: it is not determined by genetics, race, age, or reproductive history, but rather by the frequency and magnitude of lifestyle changes in an individual woman. It is the instability of habits and dietary patterns that creates the conditions under which the body remains in a state of constant adaptation.

The most significant aspect of the pathogenesis of endometrial cancer is the dynamism it creates in the hormonal–inflammatory microenvironment of the endometrium. This dynamism is not typical of either stable physiological conditions or classic metabolic disorders.

This constant change in metabolic programs disrupts the rhythm of endometrial regeneration and enhances proliferative stimuli, reducing the possibility of forming a stable hormonal profile. Metabolic instability can therefore be considered a potential independent risk factor, particularly for women with normal or slightly elevated body mass index (BMI), for whom traditional risk prediction models may not be effective.

By introducing this concept, we can better structure the available data and understand the results of studies on insulin, hormonal, inflammatory, and adipokine mechanisms within the context of a unified adaptive process rather than as isolated phenomena.

### 3.2. Epidemiological Data and the Role of Metabolic Factors in Endometrial Cancer Risk Structure

The incidence of endometrial cancer is continuing to rise worldwide, particularly in countries with high rates of urbanization and metabolic disorders. According to estimates from GLOBOCAN (2022), there has been an annual increase in new cases ranging from 1.5% to 2.5% [36]. This reflects a general trend towards a higher proportion of patients with overweight, diabetes mellitus, and other metabolic syndrome-related conditions. This increase is often linked to the “metabolic epidemic” associated with modern lifestyles.

The association between obesity and the risk of endometrial cancer has been widely studied and considered one of the strongest in oncology. Emma J Crosbie et al. (2022) found that an increase in body mass index (BMI) of every 5 kg/m^2^ is associated with a 50% increased risk of developing EC [37]. However, this relationship is not linear, and even a moderate increase in BMI (25–29.9 kg/m^2^) has been shown to be associated with a statistically significant increase in morbidity in recent years. Therefore, the threshold value at which adipose tissue starts to have a carcinogenic effect may be lower than previously thought.

Boyoung Park (2022), Prasoona Karra (2022), and Bethina Liu (2021), et al., confirm that the risk of endometrial cancer increases not only with severe obesity but also with metabolic dysfunction, even in women with a normal or borderline body mass index (BMI) [9,38,39]. For example, women with metabolic syndrome without obesity have a disease risk comparable to that of obese patients with metabolic disorders. These findings challenge the traditional view of endometrial cancer as a “hormone-dependent obesity disease” and highlight the need for a more comprehensive approach to assessing metabolic risk factors.

An additional epidemiological aspect that deserves special attention is body weight instability [10]. Yu Jin Cho et al. (2021), in their retrospective longitudinal study, believe that weight fluctuations, especially repeated cycles of “weight loss-gain”, are accompanied by pronounced morphofunctional changes in adipose tissue [11]. These changes include the development of hypoxia, fibrosis, activation of macrophages, and increased secretion of pro-inflammatory cytokines. These processes can create a metabolic environment that predisposes carcinogenesis, even in the absence of classical obesity.

In this context, repeated fluctuations in body weight can be seen as a sensitive indicator of metabolic instability. This instability reflects the body’s impaired ability to maintain homeostasis and may lead to systemic inflammation. Considering this information during anamnesis (the collection of information about a patient’s medical history) can help improve the accuracy of cancer risk assessment. It may also be important for oncogynecological screening, particularly in women who are not obese but have signs of metabolic dysfunction.

Epidemiological data suggest that it is not just obesity that determines the risk of endometrial cancer, but also the quality of a person’s metabolic background. A combination of moderate weight gain, metabolic disorders, and chronic inflammation can create a risk phenotype that has not been fully understood.

### 3.3. Pathogenetic Mechanisms Linking Metabolic Syndrome and Endometrial Cancer

The pathogenesis of endometrial cancer in metabolic syndrome is a complex interplay of interactions between endocrine, metabolic, and inflammatory systems. Traditionally, attention has been focused on the effects of obesity, hyperinsulinemia, and hyperestrogenism [40]. Each of these factors has characteristic clinical and laboratory features: obesity is associated with changes in adipose tissue distribution and increased leptin levels; hyperinsulinemia is linked to elevated insulin and C-peptide concentrations; and hyperestrogenism involves increased estradiol levels and decreased SHBG levels. These parameters constitute the typical clinical and laboratory profile of metabolic syndrome. However, they only represent the surface manifestations of a more profound metabolic imbalance that affects the regulation of inflammation, hormone metabolism, and tissue microenvironment. (Figure 2).

Gracia Fahed et al. (2022) state that one of the main links of the metabolic syndrome is insulin resistance, which leads to chronic hyperinsulinemia [41]. Excess insulin activates the PI3K/AKT/mTOR and MAPK/ERK pathways, promoting cell proliferation and reducing endometrial cell apoptosis [12]. Hyperinsulinemia also decreases the concentration of insulin-like growth factor-binding proteins (IGFBPs), increasing free insulin-like growth factor 1 (IGF-1) levels. This further enhances the proliferative signals via the IGF-1 receptor [42]. Thus, insulin resistance creates a condition for chronic cell division, leading to both endometrial hyperplasia and transformation into a tumor. At the same time, hyperinsulinemia decreases SHBG levels, increasing free estrogen levels. These free estrogens act on the endometrium to stimulate proliferation in the presence of a relative progesterone deficiency [13,14].

However, this model does not fully explain all clinical observations. A study by Prasoona Karra et al. indicates that in women without obesity, but with signs of metabolic instability, such as insulin resistance, hyperglycemia, or dyslipidemia, similar mechanisms of signaling pathway activation occur despite a normal amount of adipose tissue.

Kyoichiro Tsuchiya et al. (2025) argued that the functional state of adipocytes, rather than their number, is crucial in triggering and sustaining proliferative and inflammatory pathways [15,16,43].

Changes in the metabolic background often occur silently or are masked, without specific clinical symptoms, and therefore are not recognized by women as a separate problem and by doctors as an ongoing, developing, pathological condition. Weight fluctuations, especially within the range of normal or slightly elevated body mass index (BMI), are considered by Victoria L. Stevens et al. (2025) to be temporary and harmless, although at the molecular level, they are accompanied by repeated cycles of catabolic and anabolic phases [17]. In the catabolic phase, fat is broken down and free fatty acids, cytokines, and oxidative stress products are released. In the anabolic phase, lipid synthesis is activated, and aromatase expression (which produces estrogen) and estrogen synthesis are increased. This alternating process creates a constantly changing hormonal and inflammatory environment, in which the endometrial tissue is exposed to continuous growth stimuli, potentially increasing the risk of neoplastic transformation.

It is worth noting that similar processes can occur in both women and men. Yu Jin Cho et al. (2021) found that cyclic metabolic fluctuations, accompanied by insulin resistance and changes in the adipokine profile, may be associated with an increased risk of prostate cancer [44]. This suggests that metabolic instability is a systemic carcinogenic factor that acts through endocrine-inflammatory mechanisms that are common to both sexes.

Additionally, metabolic instability leads to changes in adipokine secretion. Aneta Słabuszewska-Józwiak et al. (2022) found that an increase in leptin levels and a decrease in adiponectin during periods of weight gain create a pro-oncogenic microenvironment that activates the JAK/STAT and NF-κB pathways [45,46].

With a subsequent decrease in body weight, these indicators do not always return to their initial levels, indicating the continued activation of pro-inflammatory mechanisms even after normalization of body weight and prolonged adverse effects [18,47]. Therefore, endometrial carcinogenesis in metabolic syndrome should not be considered a static process. It is a dynamic system where metabolic instability plays a key role, rather than obesity, due to chronic fluctuations between inflammation, hyperinsulinemia, and hormonal changes. In this context, women with overweight but not obesity, especially those with fluctuating body weight, may represent a specific risk phenotype that traditional models of pathogenesis fail to consider.

### 3.4. Adipokines and Adipose Tissue Inflammation: From a Quantitative to a Qualitative Risk Phenotype

Modern concepts of the role of adipose tissue in endometrial cancer pathogenesis have undergone significant changes. Previously, adipose tissue was considered a passive depot responsible for determining the degree of estrogen exposure, but now it is recognized as an active endocrine and immune system organ. Adipose tissue produces a diverse range of biologically active molecules, such as adipokines and cytokines, that form a metabolic and inflammatory microenvironment promoting carcinogenesis [48,58].

The key mediators in this system are leptin and adiponectin, whose dysregulation leads to a shift from metabolic equilibrium to chronic inflammation. Leptin, which is directly correlated with the mass of adipose tissue, has mitogenic and angiogenic properties [19]. It binds to the Ob-R receptor, activating JAK/STAT, MAPK, and PI3K/AKT signaling pathways, stimulating endometrial cell proliferation and angiogenesis, while inhibiting apoptosis [49].

Moreover, leptin can induce aromatase expression, leading to increased local estrogen formation and hormonal stimulation of the endometrium. In contrast, adiponectin has an opposite effect, increasing tissue sensitivity to insulin and protecting the endometrium from proliferative hyperstimulation. This is supported by the findings of Haoxin Tina Zheng et al., who conducted a systematic review on adiponectin’s role in the endometrium [20,50].

A violation of the ratio of leptin and adiponectin (L/A ratio) has been shown to be one of the most accurate biochemical indicators of adipose tissue dysfunction. Unlike BMI, this indicator reflects the qualitative characteristics of the metabolic status, such as inflammation, oxidative stress, and the endocrine activity of adipocytes [21].

Aarthi S. Jayraj et al. (2015) [22] conducted a case–control study and found that an increased L/A ratio was associated with an increased risk of endometrial cancer, even in women who were not obese. This emphasizes the importance of using functional rather than morphometric criteria to assess adipose tissue [22].

It should be noted that the assessment of leptin levels as a risk factor for endometrial cancer is not typically performed in routine clinical practice. If it is performed, it is often limited to a single measurement, which does not allow for tracking the temporal dynamics of the indicators. We believe that regular monitoring of the L/A ratio over time is a promising method for detecting precancerous conditions and early identification of high-risk groups, particularly among women who are not obviously obese.

New adipokines, such as visfatin and resistin, also play an important role in the development of chronic inflammation. These molecules not only regulate energy metabolism but also activate inflammatory signaling pathways. Elevated levels of visfatin and resistin have been observed in women with metabolic syndrome, and they correlate with increased tumor aggression [23,51]. Adipokines form a complex network of signaling pathways in which inflammation and cell proliferation reinforce each other, leading to the development of disease.

In the context of the relationship between metabolic syndrome and endometrial cancer, adipose tissue inflammation is of particular importance as a morphological phenomenon. Adipocyte hypertrophy leads to local hypoxia, activation of macrophages, and the formation of characteristic structures known as “crown-like structures” (CLS). Dhruvi Lathigara et al. described these structures as clusters of immune cells surrounding necrotizing adipocytes in 2023 [52].

These sites act as an active source of inflammatory cytokines such as IL-6, TNF-α, and MCP-1, which promote systemic inflammation and stimulate proliferation in the endometrium [24]. Charlotte Vaysse et al. conducted a case–control study in 2017 involving 107 women and found that CLS can be present not only in severely obese individuals, but also in those with moderate metabolic disorders and fluctuating body weight [25]. This confirms that the inflammatory activity of adipose tissue can remain high even with a normal body mass index (BMI) if the balance between lipogenesis and lipolysis is disrupted.

At the same time, adipose tissue in different anatomical areas is not metabolically identical. Visceral and subcutaneous fat deposits differ in their sensitivity to lipolysis, the rate of remodeling, and the composition of adipokines and cytokines. In women, these differences are particularly pronounced during periods of fluctuating body weight. Some areas, such as the abdomen and thighs, undergo more active lipolysis while others maintain a stable amount of adipose tissue. This leads to the formation of local foci of chronic inflammation where the processes of fat cell destruction and regeneration are asynchronous, creating a microenvironment that produces increased levels of pro-inflammatory factors.

This feature is also of practical importance. In an attempt to improve local body fat distribution, women often turn to anti-cellulite massages, physiotherapy, local injections, or hardware techniques without considering their overall metabolic status. Intervening in adipose tissue while a latent metabolic imbalance is present can increase local inflammation, disrupt microcirculation, and activate macrophages, leading to chronic subclinical inflammation. As a result, external aesthetic interventions on adipose tissue may inadvertently exacerbate the same pathophysiological process that underpins metabolically associated carcinogenesis.

This concept shifts the focus of research from the quantification of adipose tissue to its functional state and local inflammatory changes. Rather than the volume of fat mass, it is the endocrine and immune activity of adipocytes that determines the interaction between the metabolic syndrome and carcinogenesis. Additionally, there is a hypothesis that in women with frequent weight fluctuations, inflammation in adipose tissue does not subside but rather becomes more pronounced due to the reactivation of macrophages and remodeling of the extracellular matrix.

Thus, adipokine imbalance and persistent inflammation of adipose tissue should be considered as a possible mechanism linking metabolic syndrome and endometrial cancer, even in the absence of obesity. This relationship may be especially significant in conditions of metabolic and hormonal instability, as we have seen. Therefore, future research should focus on understanding the functional and morphological characteristics of adipose tissue, both systemically and locally, and exploring their potential as predictors of cancer risk in women.

### 3.5. The Role of Sex Hormones and SHBG in the Pathogenesis of Endometrial Cancer: Stable Hyperestrogenism Versus Hormonal Fluctuations

Hormonal disorders are traditionally considered a key link in the pathogenesis of endometrial cancer. Hyperestrogenism in conditions of relative progesterone deficiency creates conditions for chronic stimulation of the endometrium and its neoplastic transformation [53].

However, a retrospective study by Xiao et al. (2021) suggests that in women with metabolic syndrome, hormonal changes are more complex than simply an increase in estrogen levels. These changes include dynamic, fluctuating patterns of hormonal regulation, influenced by factors such as body weight, adipose tissue metabolism, and inflammation [26].

The main source of estrogen in postmenopausal women is the peripheral aromatization of androgens in adipose tissue. This process is increased by the activity of pro-inflammatory cytokines, such as IL-6 and TNF-α, which induce the expression of aromatase and leptin. These factors stimulate the local synthesis of estrogen. In women with metabolic syndrome, this process is more pronounced, leading to higher levels of estradiol and estrone.

However, in women without obesity but with metabolic dysfunction, the volume of adipose tissue may be moderate. However, its enzymatic activity and endocrine density are significantly higher, making even a small amount of metabolically active adipose tissue create a hormonal background similar to obesity.

The key factor in this background is SHBG. Hyperinsulinemia, a characteristic of the metabolic syndrome, inhibits its synthesis in the liver, leading to an increase in free, biologically active estrogen and androgen levels [27]. As a result, even with a modest increase in total steroid hormone levels, their bioavailability to the endometrium significantly increases. The endometrium is therefore constantly exposed to high levels of steroid hormones that are not balanced by progesterone.

In women with significant or frequent weight fluctuations, the metabolic state can change dramatically: from periods of insulin resistance and high estrogen levels with weight gain, to periods of relatively low estrogen levels with weight loss.

The measurement of relevant hormones has been well established and is commonly used in clinical practice, mainly for the diagnosis and monitoring of endometrial cancer risk. Additionally, it is known that these metabolic changes are accompanied by fluctuations in the levels of leptin, adiponectin, and SHBG, creating an unstable hormonal pattern that the endometrium responds to with repeated cycles of growth, inflammation, and repair [28,29,30].

However, in laboratory studies, only a single value of the indicators is usually recorded, linked to the date and time of sampling. The concentrations of the metabolic and hormonal markers listed can vary significantly even within a few hours or days, and do not correspond with subsequent measurements. So far, there have been very few studies with prolonged dynamic assessment based on daily or multi-day monitoring. Recording these fluctuations could serve as direct confirmation of metabolic instability in the body, similar to Holter monitoring, which detects episodes of arrhythmia during the day.

The clinical feature of this situation is its hidden and erased nature. Women and doctors often interpret symptoms, such as cycle disorders, mastalgia, libido swings, and body weight fluctuations, as a result of functional changes in the ovaries. However, these symptoms are actually caused by the activity of adipose tissue, which is an important endocrine organ that actively regulates steroidogenesis.

Adipose tissue not only serves as a source for the aromatization of androgens into estrogens, but it also participates in feedback mechanisms that regulate the level of sex hormones. Fluctuations in the volume of adipose tissue directly affect the levels of sex hormones, with more intense lipolysis leading to the release of more steroid precursors, and vice versa. Therefore, adipose tissue not only reflects hormonal fluctuations but also forms them.

Special attention should be given to the phenomenon of relative hyperandrogenism, which occurs when the levels of estrogens and progesterone decrease, while the concentrations of testosterone and dihydrotestosterone remain the same or decrease slightly in women [31]. Under these circumstances, an imbalance occurs between androgenic and estrogenic effects, leading to increased proliferation, impaired differentiation, and increased risk of neoplastic transformation of endometrial and cervical epithelial cells [54].

From a pathophysiological perspective, this condition can be seen as hormonal “fluctuations” involving estradiol, progesterone, testosterone, and dihydrotestosterone. These hormones’ alternating peaks and valleys create a chronically unstable endocrine environment that leads to the accumulation of cellular errors in the endometrium. Additionally, testosterone, with its catabolic effects, enhances lipolysis and the destruction of adipose tissue. This process releases lipid mediators and cytokines, which promote inflammation. As a result, a self-perpetuating pathological cycle forms, where hormonal dysregulation, including fluctuations in estradiol, progesterone, and androgens, and changes in sex hormone-binding globulin (SHBG) concentrations, initiate metabolic instability. The latter is manifested by lipid metabolism disorders and increased lipolysis. This secondary affects the secretion of sex hormones, leading to a decrease in their levels. Additionally, it increases the production of pro-inflammatory cytokines and mediators, causing inflammation and an endocrine–metabolic imbalance.

This inflammation, combined with the endocrine–metabolic imbalance, leads to chronic and irregular stimulation of the endometrial cells, reducing their programmed cell death ability. This accumulation of cellular errors and DNA damage supports antigenic stimulation, resulting in inflammatory activity and hormonal reactivity of the endometrium. Consequently, a feedback loop is initiated, reinforcing the original disorder and gradually transforming the system into a precancerous state.

A similar mechanism can be observed in gastric carcinogenesis, where a decrease in hydrochloric acid secretion increases the risk of cancer. Atrophy also increases this risk, and each stage of pathology becomes a background for the next one.

Additionally, the resumption of a high-fat diet activates lipogenesis and hormonal production, stimulating cell proliferation. This stimulation often exceeds baseline levels, and the endometrium does not have enough time to adapt structurally or functionally. This creates conditions for dysplastic processes to progress. This mechanism can be clearly seen in Figure 3.

Such erratic changes in the hormonal profile can be more dangerous than chronic hyperestrogenism in obesity, as they stimulate unstable endometrial regeneration processes and increase the likelihood of genetic errors during cell division. Additionally, the alternation between states of “hormonal excess” and “reduction” increases the sensitivity of the endometrial receptor system, creating conditions for hyperreactivity even with moderate changes in hormone levels.

The imbalance between estrogen and progesterone is exacerbated by a decrease in progesterone receptor expression (PR), which occurs in metabolic syndrome and chronic inflammation. This makes the endometrium resistant to the anti-proliferative effects of progesterone, turning estrogen stimulation into an uncontrolled proliferation impulse.

Thus, in the development of endometrial cancer in women with metabolic syndrome, hormonal instability, and not just hyperestrogenism, may be the key factor. Fluctuations in weight, insulin resistance, and chronic inflammation create a unique phenotype characterized by hormonal and metabolic instability, which supports the cyclic activation of the endometrium and contributes to carcinogenesis.

### 3.6. Clinical and Morphological Features of Endometrial Cancer in Metabolic Syndrome: The Impact of Metabolic Instability

The clinical course and morphological characteristics of endometrial cancer in women with metabolic syndrome show a wide range of differences from the traditional concept of the “hormone-dependent” type of the disease. It is traditionally assumed that metabolic disorders contribute to the development of well-differentiated type I endometrioid tumors that progress relatively favorably. However, a retrospective study by Yuanpei Wang et al. (2024) suggests a more complex picture: metabolic abnormalities alter the biology of the tumor, making it more aggressive even with a typical endometrioid appearance [4,41].

Yuman Wu et al. (2025), in their retrospective study, showed that women with metabolic syndrome are more likely to have tumors with deeper myometrial invasion, lymph node involvement, and a tendency towards recurrence [32]. This confirms that the combination of metabolic and inflammatory disorders alters the biology of the tumor, making it more aggressive, even within the context of the traditionally “estrogen-driven” type of endometrial cancer. These features are observed not only in severe obesity, but also in overweight, especially when combined with signs of insulin resistance or impaired glucose tolerance [33]. As a result, the quality of the metabolic profile appears to be a more significant predictor of tumor aggressiveness than body mass index alone.

An important pathogenic component that determines the morphological features of the tumor is the inflammatory microenvironment [55]. In metabolic syndrome, this is formed through the activation of macrophages and the release of IL-6, TNF-α, and other cytokines.

Special attention should be paid to a group of women with metabolic instability, which is characterized by recurring fluctuations in body weight, changes in insulin, leptin, and sex hormone levels. C. M. Nagle et al. (2013), in a retrospective case–control study, showed that endometrial cancer often develops in these patients at a younger age and has a hybrid morphological phenotype, combining features of both endometrioid and serous tumors [34]. It can be assumed that the constant metabolic and hormonal fluctuations in these women create conditions for genetic instability in endometrial cells, disrupting DNA repair and epigenetic regulation processes.

Fluctuations in weight are especially common during times of severe physiological and emotional stress. After pregnancy, particularly if it ends with any type of termination—childbirth, abortion, or miscarriage—the body undergoes a drastic hormonal change accompanied by alterations in lipid and carbohydrate metabolism [56]. Attempts to lose weight quickly after childbirth through strict restrictions and crash diets often lead to metabolic stress and subsequent episodes of overeating. These “pendulum” cycles of weight loss and subsequent fat accumulation disrupt the adaptation phase (“plateaus”) needed to stabilize hormonal regulation.

During such periods, the endocrine system does not have enough time to recover: the uterus and endometrium undergo repeated waves of stimulation and hormonal deficiencies, and the glands in the cervix and the endometrial epithelium become particularly sensitive to changes in estrogen and androgen levels. Additionally, stress, both physical and emotional, can trigger this process, as it leads to an increase in cortisol levels [57]. Chronically high cortisol levels contribute to an increase in cholesterol and low-density lipoprotein levels, creating a pool of lipids available for the production of sex hormones when there are disruptions in diet or hormonal imbalances.

Wanyu Yang et al. (2023) conducted a prospective case–control study and found that thyroid hypofunction can be a particularly unfavorable condition for metabolic processes. This condition slows down metabolic reactions, reduces lipolysis rates, and increases insulin levels [35]. Together, these factors can lead to a chronic metabolic overload that exposes the endometrium to an unstable hormonal rhythm.

In terms of clinical aspects, metabolic instability not only affects the risk of disease occurrence but also influences the course of the illness. Patients with this condition are more likely to experience abnormal hormonal responses to treatment, less predictable outcomes from progestogen therapy, and an increased frequency of relapse after surgical intervention.

A number of retrospective case–control studies have demonstrated that unstable fluctuations in body weight following treatment are linked to worse overall and disease-free survival outcomes [26,33]. This emphasizes the significance of dynamic metabolic management as a prognostic indicator.

Thus, the clinical and morphological features of endometrial cancer in metabolic syndrome reflect a dynamic disturbance of metabolic and hormonal regulation, rather than a static one. In women without obesity, but with signs of metabolic lability, the disease can be as aggressive as in those with severe obesity. This highlights the need to re-evaluate existing diagnostic approaches. It is important to consider not only individual indicators of body weight, glycemic levels, and other parameters, but also their variability over short, medium, and long periods of time, as an independent prognostic factor, especially in individuals who experience intermittent weight changes due to personal factors or external interventions.

Modern concepts of the pathogenesis of endometrial cancer associated with metabolic syndrome do not fully reflect the role of “metabolic instability”, a chronic process of unnatural and chaotic transition between different homeostatic states. To address this, it is promising to develop clinical recommendations aimed at stabilizing metabolic profiles. These recommendations could include controlling body weight, adopting a healthy lifestyle, and, in some cases, using pharmacological support such as hormonal contraception to regulate hormonal fluctuations. Such strategies can provide a “metabolic pause” for the body, reducing the amplitude of fluctuations and creating a more stable environment for maintaining homeostasis and cellular division processes in organs and systems, including the endometrium.

However, randomized clinical trials are necessary to implement these approaches, as the current data is insufficient for meta-analysis and do not allow for a comprehensive assessment of endometrial carcinogenesis within the context of dynamically changing metabolic states.

## 4. Discussion

This discussion aims to interpret the findings of the present review in the context of current evidence on metabolic instability and the pathogenesis of endometrial cancer.

The traditional view of endometrial cancer being linked to obesity and hyperestrogenism has long been widely accepted. Adipose tissue has been seen primarily as a source of estrogen, creating a hormonal environment that is favorable for endometrial growth [40]. However, recent data suggest that this model only represents one aspect of the disease’s pathogenesis.

Endometrial cancer is not simply a result of excess body weight; it is a manifestation of metabolic imbalance, with the amplitude and frequency of fluctuations in metabolism being more significant than the severity of metabolic disturbances [7,9,10,11,38,39].

Modern concepts are shifting the focus away from a static understanding of the metabolic syndrome and towards a dynamic concept. Instead of focusing on the level of body weight, glucose, insulin, and other factors, the variability of these factors becomes crucial. Fluctuations in metabolic processes rather than persistent disturbances form a state of chronic cellular stress and inflammation, which creates a favorable microenvironment for neoplastic changes in the endometrium [15,16,43].

In this context, repeated fluctuations in body weight can be seen as a sensitive marker of metabolic instability. These fluctuations reflect the body’s tendency towards impaired homeostasis and systemic inflammation. Considering this information when collecting anamnesis can help improve the accuracy of cancer risk assessment and is important for oncogynecological screening. This is especially true for women who are not obese but have signs of metabolic dysfunction.

The pathogenesis of endometrial cancer in the context of metabolic syndrome is a complex interplay between the endocrine, metabolic, and immune systems. Obesity, hyperinsulinemia, and hyperestrogenism are commonly considered to be the main components of this process, each with its own clinical and laboratory manifestations [2,3,4,5,12,13,14,38,41,42]. However, these factors only reflect the surface manifestations of more subtle and dynamic regulatory mechanisms, where transient states play a crucial role. These transient states are characterized by short but frequent episodes of metabolic dysregulation.

One of the main mechanisms is the alternating process of catabolism and anabolism, which occurs with changes in body weight. During catabolism, lipolysis, the release of free fatty acids, cytokines, and oxidative products, is increased, while during anabolism, lipogenesis, estrogen synthesis, and androgen aromatization are activated [17]. This creates a hormonal and inflammatory cycle, where the endometrium experiences repeated waves of stimulation and inflammation.

In adipose tissue, these processes manifest morphologically as hypoxia, macrophage activation, and the formation of “crown-like structures” (CLS). These are clusters of immune cells surrounding necrotizing adipocytes [24,52]. CLS is seen not only in severe obesity but also in women with mild metabolic disorders and fluctuating weight, indicating that adipose tissue inflammation can be high even at a normal body mass index (BMI) [25].

In addition, adipose tissue from different anatomical zones has a different metabolic profile [13,38]. During periods of weight fluctuation in women, visceral and subcutaneous fat deposits respond differently—some shrink actively, while others remain stable. This asynchronous response creates local foci of inflammation and metabolic activity, creating a microenvironment that supports neoplasia [13,38,39].

Another factor is external interventions, such as anti-cellulite massages and physiotherapy, without considering the metabolic status of the patient. These treatments can increase inflammation and tissue remodeling in adipose tissue that is already undergoing metabolic instability, unintentionally enhancing the pathogenic mechanisms of metabolism-related carcinogenesis.

Hormonal instability plays an equally important role in the development of endometrial cancer. Fluctuations in body weight lead to alternating phases of elevated insulin levels, estrogen levels, and decreased estrogen levels. This is accompanied by impaired synthesis of sex hormone-binding globulin (SHBG) and changes in the sensitivity of endometrial receptors [13,14,27,28,29,30].

The combination of these factors creates an unstable hormonal and metabolic profile. The endometrium experiences repeated waves of stimulation and regeneration, followed by depletion of regulatory mechanisms and the accumulation of replication errors. In contrast to stable hyperestrogenism observed in obesity, these fluctuations create an unpredictable hormonal environment that leads to increased oxidative stress and a higher likelihood of replication errors.

Of particular importance is the phenomenon of relative hyperandrogenism, which occurs when estrogen and progesterone levels decrease [31]. Under these conditions, continued exposure to testosterone and dihydrotestosterone can lead to excessive proliferation of endometrial and cervical cells. This can also support adipose tissue breakdown, increasing inflammation [54].

Metabolic instability often occurs during periods of physiological stress, such as after pregnancy, abortion, childbirth, or sudden changes in diet. Rapid postpartum weight loss, emotional breakdowns, and stress-related overeating can lead to cycles of hormonal and metabolic fluctuations that disrupt adaptation phases (known as “plateaus”) [56,57]. During this time, the endocrine system is unable to fully recover, and target organs such as the uterus and endometrium are affected by fluctuating hormonal levels.

Chronic stress and low levels of cortisol can also contribute to hyperinsulinemia and hyperlipidemia. Hypothyroidism can further slow down metabolism and reduce the body’s ability to recover [35].

Thus, endometrial cancer should not be considered a disease of obesity, but rather a disease of metabolic dysregulation, which is characterized by a loss of the body’s ability to maintain homeostasis and frequent fluctuations in metabolic parameters. Variations in body weight, blood sugar levels, and hormone levels are becoming an important predictor of risk for endometrial cancer, rather than simply a contributing factor.

This has important implications for prevention and early detection. The measurement of metabolic variability over time, such as fluctuations in body weight, insulin levels, glucose levels, leptin levels, and adiponectin levels, may provide a more accurate tool for stratifying risk of endometrial cancer than traditional static measures.

The introduction of the concept of “metabolic instability” provides a fresh perspective on the relationship between metabolism and oncological processes. Instead of focusing on absolute values, this approach focuses on the dynamic characteristics of metabolism, which helps to clarify the pathogenetic model of endometrial cancer. This shift in focus also opens up opportunities for the development of personalized monitoring, prevention, and intervention strategies for women at high risk for the disease, allowing for a more targeted approach to metabolic fluctuations.

### 4.1. Clinical Implications and Future Directions

From a clinical standpoint, the findings underscore the importance of incorporating markers of metabolic variability into endometrial cancer risk stratification models and developing preventive strategies aimed at stabilizing metabolic homeostasis in non-obese women.

### 4.2. Limitations

This review has certain limitations. The included studies were heterogeneous in design, methodology, and population characteristics. Definitions of metabolic variability were inconsistent, and few studies applied standardized criteria for assessing metabolic instability. Furthermore, most of the available data were retrospective, limiting the ability to establish causal relationships. Despite these limitations, the studies showed consistent trends linking metabolic instability with endometrial carcinogenesis. Therefore, the overall certainty of the evidence was assessed as moderate, reflecting methodological heterogeneity and limited longitudinal data. These limitations should be considered when interpreting the findings and in designing future prospective investigations.

A potential reporting bias cannot be excluded, as studies with negative or non-significant findings may be underrepresented in the literature. Despite these limitations, the studies demonstrated consistent trends linking metabolic instability with endometrial carcinogenesis. Therefore, the overall certainty of the evidence was assessed as moderate, reflecting methodological heterogeneity, limited longitudinal data, and potential publication bias. These limitations should be considered when interpreting the findings and in designing future prospective investigations.

## 5. Conclusions

An analysis of current data has revealed that the etiology of endometrial cancer is not solely limited to the classical model based on obesity and persistent hyperestrogenism. Instead, the disease often develops in women with normal or only moderately elevated body mass index (BMI), especially if they experience a phenomenon of metabolic instability—constant chaotic fluctuations in body weight, insulin levels, glucose levels, adipokine levels, and steroid hormone levels.

Unlike stable metabolic disorders, it is the variability of these parameters that creates a chronic inflammatory and proliferative microenvironment in the endometrium. Frequent changes between anabolic and catabolic states lead to the activation of hormonal and signaling pathways, disruption of DNA repair, and accumulation of cellular errors.

The key carcinogenic link is the functional state of adipose tissue and its transitional forms with an unstable composition. Even with its small volume, this tissue can develop inflammatory changes and alter the secretion of leptin and adiponectin. It also enhances aromatase activity locally, making adipose tissue an active endocrine–immune organ.

Patients with metabolic instability often have more aggressive morphological and clinical features of the tumor. This emphasizes the need for a more accurate approach to risk assessment. Individual indicators such as BMI and hormone levels are not enough. A more precise criterion for assessing a woman’s homeostasis would be to track changes in metabolic parameters over time. Analyzing metabolic instability studies is a promising area of research.

Therefore, the term “metabolic instability” exists and has clinical significance as a syndrome, and should be considered a new independent risk factor for endometrial cancer. It should be taken into account in the early diagnosis, prognosis, and development of prevention strategies.

## Figures and Tables

**Figure 1 cancers-17-03840-f001:**
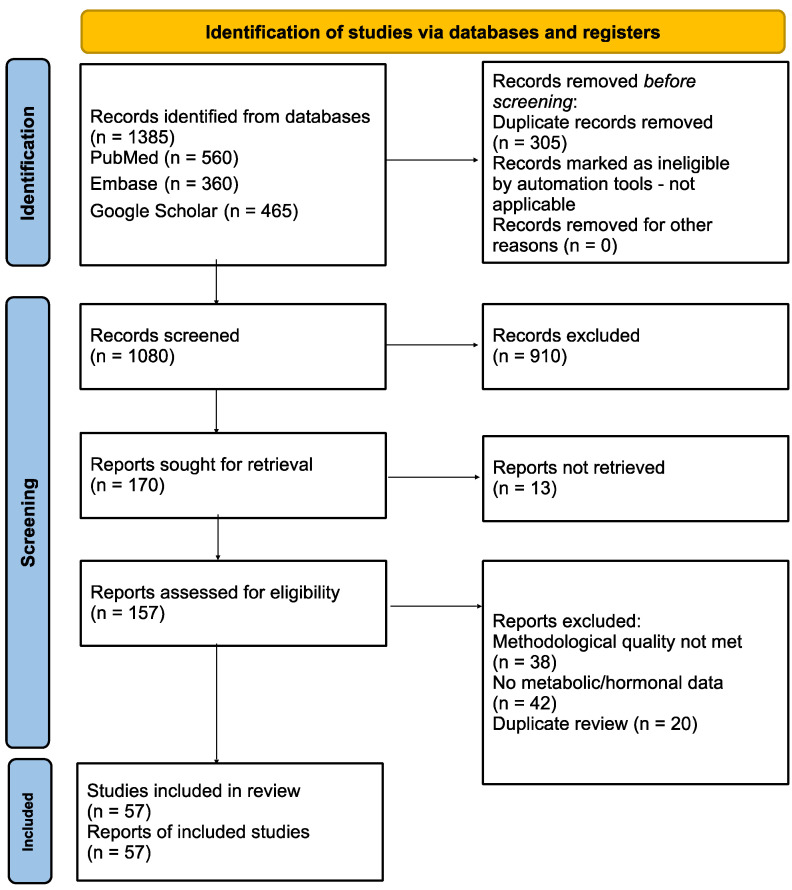
A block diagram of research selection based on the PRISMA principle. The figure shows the stages of literature selection, screening, acceptance assessment, and inclusion in the program in accordance with the PRISMA 2020 guidelines [8].

**Figure 2 cancers-17-03840-f002:**
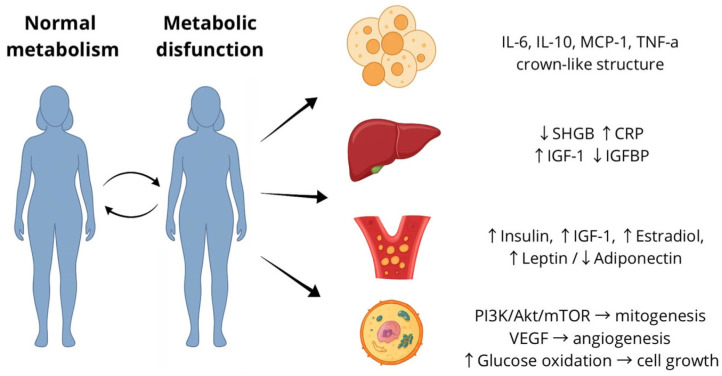
Conceptual model of metabolic dysfunction and its systemic effects contributing to endometrial carcinogenesis. Cyclic transitions between normal metabolism and metabolic dysfunction trigger inflammatory cytokine release (IL-6, IL-10, MCP-1, TNF-α), altered hepatic signaling (↓ SHBG, ↑ CRP, ↑ IGF-1, ↓ IGFBP), hormonal imbalance (↑ insulin, ↑ estradiol, ↑ leptin, ↓ adiponectin), and activation of intracellular pathways (PI3K/Akt/mTOR, VEGF), leading to mitogenesis, angiogenesis, and enhanced cellular proliferation.

**Figure 3 cancers-17-03840-f003:**
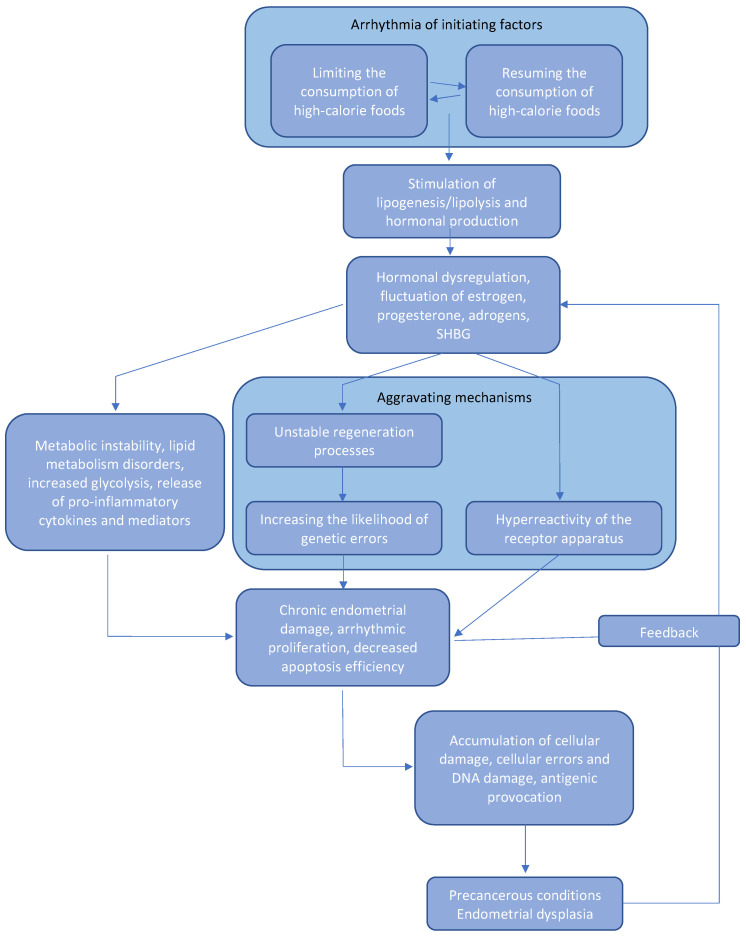
The pathological cycle of metabolic instability.

**Table 1 cancers-17-03840-t001:** Clinical trials included in the article.

Author	Year of Publication	Country	Research Design	Sample Size
Tong Y., et al. [6]	2025	China	Two-sample Mendelian randomization (MR) analysis with multivariable adjustments, leveraging data from IEU OpenGWAS	
Park B., et al. [9]	2022	East Asia	Representative study	6,097,686
Lin X.Y., et al. [10]	2025	China	Cohort Study	20,056
Cho Y.J., et al. [11]	2021	Korea	Cohort Study	1,759,848
Wang Y., et al. [12]	2024	China	Cohort Study	406
Arthur R.S., et al. [13]	2021	UK	Cohort Study	149,928
Balogh Z., et al. [14]	2025	Hungary	A cross-sectional study	195
Wada E., et al. [15]	2025	Japan	Experimental study on mice	
Reinisch I., et al. [16]	2025	Switzerland	Cohort Study	77
Stevens V.L., et al. [17]	2025	USA	Cohort Study	45,004
Caslin H.L., et al. [18]	2023	USA	Experimental study on mice	
Boroń D., et al. [19]	2021	Poland	Case–control study	60
Zhang Z., et al. [20]	2022	China	Cohort Study	106
Vatannejad A., et al. [21]	2023	Iran	Case–control study	200
Jayraj A.S., et al. [22]	2025	India	Case–control study	80
Zhu X., et al. [23]	2023	China	Case–control study	219
Moukarzel L.A., et al. [24]	2022	USA	Cohort Study	101
Vaysse C., et al. [25]	2017	Norway	Case–control study	107
Yang X., et al. [26]	2021	China	Case–control study	506
Zhang H., et al. [27]	2024	China	A cross-sectional study	206
Komaroff M. [28]	2016	USA	Case–control study	158
Welti L.M., et al. [29]	2017	USA	Observational Study	80,943
Gonzalo-Encabo P., et al. [30]	2021	Canada	Randomized Controlled Trial	400
Mullee A., et al. [31]	2021	UK	Observational Study	159,702
Wu Y., et al. [32]	2025	China	Case–control study	109
Li X., et al. [33]	2021	China	Cohort Study	111
Nagle C.M., et al. [34]	2013	Australia	Case–control study	2936
Yang W., et al. [35]	2023	China	Case–control study	1243

**Table 2 cancers-17-03840-t002:** Reviews included in the article.

Author	Year of Publication	Review Design	Purpose of the Review
Crosbie E.J., et al. [1]	2022	Review	Understanding the molecular biology of EC.
Wang L, et al. [2]	2020	Systematic review and meta-analysis	Confirm the association of metabolic syndrome with the risk of endometrial cancer.
Zhao S., et al. [3]	2022	Review	Description of endometrial cancer associated with Lynch syndrome.
Zheng W., et al. [4]	2025	Review	In this article, the effects of the gut microbiota and its metabolites on the occurrence and development of endometrial cancer is reviewed.
Scherübl H., et al. [5]	2022	Review	Description of the relationship between metabolic syndrome and cancer risk.
de Andrade Mesquita L., et al. [7]	2023	Review	Outline epidemiological and pathophysiological aspects of the relationship between obesity, diabetes, and malignancies.
Page M.J., et al. [8]	2021	Guideline for reporting systematic reviews	PRISMA 2020 abstract checklist, and the revised flow diagrams.
Bray F., et al. [36]	2024	Global cancer statistics	Global cancer statistics 2022
Lega I.C., et al. [37]	2020	Review	summarizing existing data describing the relationship between obesity, diabetes, and cancer.
Karra P., et al. [38]	2022	Review	Assessment of various definitions of metabolic dysfunction depending on the risk of obesity-related cancer and mortality after cancer diagnosis
Liu B., et al. [39]	2021	Narrative review	synthesis of existing data describing the relationship between metabolic syndrome and cancer.
Makker V., et al. [40]	2021	Review	This review provides updated information on the epidemiology, pathophysiology, diagnosis, and molecular classification of endometrial cancer.
Fahed G., et al. [41]	2022	Review	Description of Epidemiology, histopathology and pathophysiology Of Metabolic Syndrome.
Deng F., et al. [42]	2024	Meta-analysis	This meta-analysis examined the association between MetS and survival outcomes in EC patients.
Tsuchiya K., et al. [43]	2025	Review	Summarizing data on metabolically healthy obesity.
Morlacco A., et al. [44]	2022	Review	This review examines the available evidence on the relationship between metabolic syndrome and prostate cancer risk.
Słabuszewska-Jóźwiak A., et al. [45]	2022	Review	The role of leptin and adiponectin in the development of endometrial cancer is considered.
Buonaiuto R., et al. [46]	2022	Review	The role of leptin in the pathogenesis of breast cancer is considered.
Hinte L.C., et al. [47]	2024	Review	The effect of metabolic memory is considered.
Tilg H., et al. [31]	2025	Review	The role of adipokines in the development of metabolic inflammation is considered.
Bunnell B.A., et al. [48]	2022	Review	The role of adipose tissue in the pathogenesis of cancer is considered.
Ray I., et al. [49]	2022	Systematic review	The effect of adipocytokines on the development of endometrial cancer is being investigated.
Zheng H.T., et al. [50]	2024	Systematic review and meta-analysis	The influence of circulating biomarkers on the risk of endometrial cancer is assessed.
Kołakowska K., et al. [51]	2025	Review	The role of adipokines associated with obesity in ovarian cancer is considered.
Lathigara D., et al. [52]	2023	Narrative review	The narrative review of the molecular mechanisms of Western diet-induced obesity and obesity-related carcinogenesis.
Bassette E., et al. [53]	2024	Review	The etiology, risk factors and prognosis of endometrial cancer in women of reproductive age are considered.
Zhu P., et al. [54]	2025	Systematic review and meta-analysis	This meta-analysis was carried out to investigate the relation of circulating concentrations of SHBG to EC risk.
Elia I., et al. [55]	2021	Review	The metabolites and microenvironment of the tumor are considered.
Hicks L.E., et al. [56]	2025	Systematic review and meta-analysis	The postpartum mental health of women and the impact of physical activity on it are being investigated.
Wardle J., et al. [57]	2011	Review	Stress is considered as a risk factor for obesity.

## Data Availability

All data are available in the cited sources.

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
