# Peer review of "The Role of “Metabolic Instability” as a New Risk Criterion in the Pathogenesis of Endometrial Cancer"

_cancers, 2025, doi:10.3390/cancers17233840_

Round 1
Reviewer 1 Report
Comments and Suggestions for Authors
The manuscript presents a timely and conceptually ambitious systematic review exploring metabolic instability as a novel oncogenic risk factor for endometrial cancer beyond obesity. The topic is potentially significant, as it reframes traditional risk models with attention to dynamic hormonal and adipokine fluctuations. However, while the narrative demonstrates thematic depth and breadth, it suffers from severe methodological and analytical deficiencies. The absence of quantitative synthesis despite a claimed systematic approach weakens the validity of conclusions. The selection criteria and evidence grading are underdeveloped, and some core concepts remain vaguely defined or speculative. Furthermore, repetitiveness and excessive theorizing without robust data diminish the overall scientific rigor. Substantial revisions are required.
1. The methodological foundation of the systematic review is insufficiently developed, casting doubt on the reliability of conclusions. The authors claim adherence to PRISMA 2020 guidelines, but also state that “the review protocol was not prospectively registered” and that “a formal meta-analytic synthesis was not feasible.” These admissions undermine reproducibility and analytical rigor. Furthermore, the study selection process lacks essential detail: “Full-text screening identified 57 publications meeting the inclusion criteria” — yet no list of included studies, nor their basic characteristics (sample size, design type, region), is presented in tabular or visual format. The authors report thematic synthesis, but fail to provide structured data extraction matrices. Without transparency in study selection, it is impossible to assess bias, heterogeneity, or the weight of evidence across domains. The entire review collapses into narrative speculation rather than analytical synthesis.
2. The validity of the central concept “metabolic instability”, as an independent oncogenic risk factor remains poorly substantiated. The authors define it vaguely as “recurrent fluctuations in body weight, glucose, insulin, adipokines, and sex hormones,” yet operational criteria for this condition are absent. Moreover, the narrative repeatedly infers causality from correlative studies without rigorous evidence. For example, the manuscript states: “recurrent weight fluctuations may serve as a sensitive marker of metabolic instability… [and] improve the precision of individual oncological risk assessment,” but no longitudinal or interventional data are cited to support this clinical claim. The authors speculate extensively on pathophysiological pathways (“This establishes a self-reinforcing pathological loop...”) without providing quantifiable effect sizes or controlling for confounders such as age, parity, or genetic predisposition. These extrapolations exceed the limits of the evidence presented.
3. The review exhibits significant redundancy and thematic circularity. For example, the concept of weight fluctuation as a carcinogenic factor is reiterated in multiple sections (3.1, 3.2, 3.4, 3.5), often with nearly identical phrasing: “recurrent weight variability... may represent a distinct risk phenotype.” This repetition inflates word count without adding analytical depth and should be condensed.
4. Reference use is inconsistent and lacks integration. While the review cites over 50 sources, many are introduced without critical appraisal or contextual framing. For instance, statements like “Recent evidence indicates...” or “Studies have shown...” are frequently followed by references without specific study characteristics. This weakens source credibility and obfuscates the robustness of claims.
5. Certain terms lack precise definition or standardization. The phrase “metabolic memory” is used in section 3.2 and 3.3 without anchoring in established literature or mechanistic explanation. Similarly, the term “metabolic turbulence” in 3.4 appears speculative. The authors should standardize terminology and align it with endocrinological and oncological consensus definitions.
6. The sentence “Thus, current concepts regarding the relationship between metabolic syndrome and endometrial cancer require reconsideration” is overly generic and lacks scientific density. Rephrasing for clarity and precision is advised.
7. Typographical inconsistencies such as “per-formed” and “meta-analytic synthesis was not feasi-ble” (page 4) suggest inadequate proofreading and should be corrected.
Author Response
Dear reviewer,
We sincerely thank you for your careful review of our manuscript and for the valuable comments you have provided. We appreciate your input and the opportunity to improve the quality of our work. For your convenience, all changes to the text have been highlighted.
1. Comment: The methodological foundation of the systematic review is insufficiently developed, casting doubt on the reliability of conclusions. The authors claim adherence to PRISMA 2020 guidelines, but also state that “the review protocol was not prospectively registered” and that “a formal meta-analytic synthesis was not feasible.” These admissions undermine reproducibility and analytical rigor. Furthermore, the study selection process lacks essential detail: “Full-text screening identified 57 publications meeting the inclusion criteria” — yet no list of included studies, nor their basic characteristics (sample size, design type, region), is presented in tabular or visual format. The authors report thematic synthesis, but fail to provide structured data extraction matrices. Without transparency in study selection, it is impossible to assess bias, heterogeneity, or the weight of evidence across domains. The entire review collapses into narrative speculation rather than analytical synthesis.
Response: Regarding your comment about the methodological section, we have addressed your concerns and have made the necessary changes to the manuscript. The updated version includes:
- A significantly expanded methodological section that provides more detail on the search strategy, inclusion/exclusion criteria, screening process, and selection criteria.
- Clarification of the procedures for independent review and resolution of disagreements among authors.
- Information about the quality assessment tools used (AGREE II).
- Two new tables have been included with the main characteristics of all studies, both original and review. We are confident that these additions have significantly improved the transparency, reproducibility, and analytical reliability of the methodological part of the work.
These changes aim to provide a more comprehensive and transparent account of our methods, ensuring that readers can better understand the approach and rationale behind our study.
2. Comment: The validity of the central concept “metabolic instability”, as an independent oncogenic risk factor remains poorly substantiated. The authors define it vaguely as “recurrent fluctuations in body weight, glucose, insulin, adipokines, and sex hormones,” yet operational criteria for this condition are absent. Moreover, the narrative repeatedly infers causality from correlative studies without rigorous evidence. For example, the manuscript states: “recurrent weight fluctuations may serve as a sensitive marker of metabolic instability… [and] improve the precision of individual oncological risk assessment,” but no longitudinal or interventional data are cited to support this clinical claim. The authors speculate extensively on pathophysiological pathways (“This establishes a self-reinforcing pathological loop...”) without providing quantifiable effect sizes or controlling for confounders such as age, parity, or genetic predisposition. These extrapolations exceed the limits of the evidence presented.
Thank you for highlighting the need for a clearer theoretical basis for the central concept. In the revised article:
- A separate section has been added (3.1) providing a refined definition of metabolic instability, integrating clinical, metabolic, and behavioral factors.
- It is emphasized that this condition is not determined by genetic, racial, or age characteristics but is mainly formed by individual lifestyle dynamics and metabolic fluctuations.
These revisions allow for a more precise, reproducible, and clinically relevant definition of the concept.
3. Comment: The review exhibits significant redundancy and thematic circularity. For example, the concept of weight fluctuation as a carcinogenic factor is reiterated in multiple sections (3.1, 3.2, 3.4, 3.5), often with nearly identical phrasing: “recurrent weight variability... may represent a distinct risk phenotype.” This repetition inflates word count without adding analytical depth and should be condensed.
Response: Your comments were completely justified.
- The key points related to the phenomenon of weight fluctuations and metabolic instability were combined in Section 3.1.
- The text in the following sections was shortened, thematic repetitions were eliminated, and the structure was made more consistent.
We agree that these revisions have made the presentation clearer and more logical.
4. Comment: Reference use is inconsistent and lacks integration. While the review cites over 50 sources, many are introduced without critical appraisal or contextual framing. For instance, statements like “Recent evidence indicates...” or “Studies have shown...” are frequently followed by references without specific study characteristics. This weakens source credibility and obfuscates the robustness of claims.
Response: Thank you for bringing this to our attention. In the updated version:
- Brief descriptions of the studies used were added to the main conclusions.
- Links were given a uniform structure.
- Duplicate links were removed.
In our opinion, these changes have improved the quality of the argument and the transparency of the presented data.
5. Comment: Certain terms lack precise definition or standardization. The phrase “metabolic memory” is used in section 3.2 and 3.3 without anchoring in established literature or mechanistic explanation. Similarly, the term “metabolic turbulence” in 3.4 appears speculative. The authors should standardize terminology and align it with endocrinological and oncological consensus definitions.
Response: We agree that the use of terms that do not have a generally accepted clinical definition can create the impression of speculation. Based on your recommendation, we have made the following changes:
- Expressions such as "metabolic turbulence" and "metabolic memory", as well as similar terms, have been deleted or replaced with accurate and widely accepted formulations.
- The terminology has been aligned with modern endocrinological and oncogynecological standards. This work has made the text more scientific and precise.
6. Comment: The sentence “Thus, current concepts regarding the relationship between metabolic syndrome and endometrial cancer require reconsideration” is overly generic and lacks scientific density. Rephrasing for clarity and precision is advised.
Response: Regarding the lack of specificity in the conclusions about the need for revision of existing concepts, we accept your comment. In the revised manuscript, we have replaced this wording with a more substantial thesis that:
- Specifies aspects that need clarification in modern models of pathogenesis.
- Outlines the limitations of current approaches.
- Emphasizes the importance of considering the temporal variability of metabolic parameters.
Thus, the conclusion has become more accurate and focused on clinical application.
7. Comment: Typographical inconsistencies such as “per-formed” and “meta-analytic synthesis was not feasi-ble” (page 4) suggest inadequate proofreading and should be corrected.
Response: Thank you for your attention. We have carefully proofread the text and corrected all typos, which has improved the readability of the manuscript.
Additionally, we have made the following adjustments:
- The title of the article has been revised to better reflect the main idea.
- The conclusion has been rewritten to strengthen the connection between the results and the conclusions, and to emphasize their clinical relevance.
We appreciate your detailed and constructive feedback on our review. Your suggestions have helped us to improve the scientific rigor, structure, and clarity of the paper. We appreciate any further recommendations you may have for future revisions.
Kind regards,
Maria Sukhanova
(on behalf of all co-authors)
Reviewer 2 Report
Comments and Suggestions for Authors
In this review by Sukhanova et al, the authors have performed extensive literature survey to determine the risk factors for endometrial cancer
The article highlights that in addition to obesity, changes in metabolic factors and weight can act as potent factors for predicting the incidence of endometrial cancer
Author Response
Dear Reviewer,
We would like to thank you for taking the time to read our article and for your constructive feedback regarding the novel approach we have taken in considering metabolic fluctuations and dynamic factors as potential predictors of endometrial cancer risk in addition to obesity.
Based on your comments, we have revised the visual aspect of the manuscript. For your convenience, all changes to the text have been highlighted.
Specifically, we have added a new figure that clearly illustrates the pathological cycle of metabolic instability and its role in the development of precancerous changes in the endometrium. This figure helps to better understand and visualize the mechanisms discussed in the manuscript.
Thank you again for your feedback and we appreciate any further recommendations you may have as we continue to refine the manuscript.
Kind regards,
Maria Sukhanova
(on behalf of all co-authors)
Reviewer 3 Report
Comments and Suggestions for Authors
This is a study assesing the relationship between endometrial cancer and metabolic syndrome extends beyond obesity and hyperestrogenism. The authors run a systematic review that showed that metabolic instability fluctuations in body weight, glycemia, and hormone
levels that create a state of chronic cellular stress and subclinical inflammation, even at
normal BMI. All these are very interesting and unique but the article is very difficult to read and understant. A recommend a major revision not for the content but the language and quality if presentation mainly in results section.
The manuscript needs corrections as comes to coherence and eloquence.
Author Response
Dear Reviewer,
Thank you for taking the time to review our manuscript and for your thoughtful comments on the language and presentation quality. We appreciate your feedback, and we agree that there were areas where additional editing and revision were needed. For your convenience, all changes to the text have been highlighted.
Based on your suggestions, we have made the following changes:
- We have completed a thorough linguistic edit to the English text, improving readability, simplifying complex constructions and eliminating ambiguous formulations.
- The Materials and Methods, Results, and Conclusion ssections have been restructured to make the presentation more clear, consistent, and logical.
We would like to thank you again for bringing these issues to our attention. Your comments have helped us to improve the presentation of our results and overall clarity of our article. We hope you will find the revised manuscript more appropriate for publication.
If you have any further recommendations, we would be happy to consider them in our future work on the manuscript.
Kind regards,
Maria Sukhanova
(on behalf of all co-authors)
Reviewer 4 Report
Comments and Suggestions for Authors
The authors have systematically reviewed this article and the literatures are upto date and accurate. The title may be modified for better understanding. This title conveys that obesity is the only or primary diagnosis for Endometrial cancer.
- Since all the authors from the same affiliation the superscript of author numbering can be removed and the affiliation address is not given fully, the abbreviation (FSAEI HE I.M. Sechenov First MSMU of MOH of Russia) may be explained
- All the 9 authors from the same Department or laboratory or School?
- Obesity also one of the metabolic disorders then why authors have mentioned beyond obesity in the title, the other main cause is “increased level of estrogen in the body, particularly when it is not balanced by progesterone”
- “Role of Metabolic Instability” is a broad term and authors should have taken a specific metabolic instability related to Endocrine system or immune system or reproductive system which was not considered more or studies lesser
- The inclusion and exclusion criteria could have been more defined
- How often patients with endometrial cancer have been clinical studied for altered leptin/adiponectin secretion?
- The authors list the following conditions are the main cause of Endometrial cancer “chronic subclinical inflammation, altered leptin/adiponectin secretion, decreased sex hormone–binding globulin, and increased estrogen bioavailability” these are well known factor and these hormone assays are done regularly in the clinics for EC.
- In the methodology sections authors have mentioned that 57 articles included for this review but in the reference 58 articles are listed
- There are many places typographical error with hyphenated words (methodolo-gy, per-formed, in-flammation, indi-cating, variabil-ity, met-abolic, carcinogene-sis, reflect-ing, prospec-tive, limita-tions, as-sessed….many more to go, kindly check? Was it AI generated?
- The conclusion may be correlated to the title of the article along with the primary objectives of the article
- Authors stated that no funds required but in the authors contribution it mentioned that “A.F.; funding acquisition” Hope all the authors have contributed equally to this review article
Author Response
Dear Reviewer,
Thank you for your careful consideration of our work and for your informative, constructive comments. Each of them allowed us to clarify the key points of the review, improve the structure and improve the quality of the presentation of the material. Below are the detailed answers point by point. For your convenience, all changes to the text have been highlighted.
1. Comment: Since all the authors from the same affiliation the superscript of author numbering can be removed and the affiliation address is not given fully, the abbreviation (FSAEI HE I.M. Sechenov First MSMU of MOH of Russia) may be explained
Response: Thanks for the comment. In the updated version of the manuscript, we provided the full name of the organization without abbreviations, which eliminated possible ambiguities. The superscript has been removed as redundant, since all authors have the same affiliation.
2. Comment: All the 9 authors from the same Department or laboratory or School?
Response: Yes, it is. All the authors belong to the same department of Sechenov University.
3. Comment: Obesity also one of the metabolic disorders then why authors have mentioned beyond obesity in the title, the other main cause is “increased level of estrogen in the body, particularly when it is not balanced by progesterone”
Response: Thank you for your comment. We agree that the previous name did not fully reflect the content of the study. It has been replaced by:
"The role of metabolic instability as a new risk criterion in the pathogenesis of endometrial cancer," which more accurately conveys the focus of the work and does not create a false contrast to obesity.
4. Comment: “Role of Metabolic Instability” is a broad term and authors should have taken a specific metabolic instability related to Endocrine system or immune system or reproductive system which was not considered more or studies lesser
Response: Your comment has allowed us to significantly improve the structure and conceptual accuracy of the review. In the updated version:
- A separate subsection 3.1 is included with the definition of metabolic instability;
It is emphasized that this is a complex condition that develops at the junction of endocrine, metabolic, inflammatory and behavioral factors.;
It is clarified that it does not refer in isolation to any one system (endocrine, immune or reproductive), but is an integral indicator of dynamic changes in metabolic status.
5. Comment: The inclusion and exclusion criteria could have been more defined
Response: Your remark is true. Deep processing has been carried out in the "Materials and Methods" section.:
- detailed inclusion and exclusion criteria;
- the stages of search, screening and selection are described;
- the procedures for independent evaluation by the authors are explained;
- added quality assessment tools (AGREE II);
- Two tables with characteristics of all included studies are presented.
These changes have made the methodology of the manuscript more transparent and reproducible.
6. Comment: How often patients with endometrial cancer have been clinical studied for altered leptin/adiponectin secretion?
Response: We appreciate the question. In fact:
- routine determination of leptin and adiponectin is not included in the standard clinical examination of patients at risk or suspected of endometrial cancer;
- Such studies are conducted sporadically and mainly within the framework of scientific projects.
In the article, we emphasized that dynamic monitoring of these markers, rather than a one-time assessment, may be a promising area for early diagnosis and risk stratification.
7. Comment: The authors list the following conditions are the main cause of Endometrial cancer “chronic subclinical inflammation, altered leptin/adiponectin secretion, decreased sex hormone–binding globulin, and increased estrogen bioavailability” these are well known factor and these hormone assays are done regularly in the clinics for EC.
Response: We agree that the determination of sex hormones is a common practice. However:
- These indicators are usually estimated without tracking their variability over time.;
- We emphasize the importance of dynamic monitoring, which can reflect the formation of precancerous changes.
This aspect was clarified and reinforced in the final version of the manuscript.
8. Comment: In the methodology sections authors have mentioned that 57 articles included for this review but in the reference 58 articles are listed
Response: Thank you for your attentive comment. The review included 57 studies that meet the selection criteria. The additional reference refers to the PRISMA methodological document, which is not included in the set of analyzed papers, but is used to describe the methodology. This is now clearly reflected in the text.
9. Comment: There are many places typographical error with hyphenated words (methodolo-gy, per-formed, in-flammation, indi-cating, variabil-ity, met-abolic, carcinogene-sis, reflect-ing, prospec-tive, limita-tions, as-sessed….many more to go, kindly check? Was it AI generated?
Response: A small number of incorrect hyphenation and formatting artifacts were indeed present. They arose as a result of automatic text splitting. In the current version:
- The entire text has been carefully proofread,
- All typographical errors and incorrect hyphenation have been eliminated,
- Formatting is fully aligned.
10. Comment: The conclusion may be correlated to the title of the article along with the primary objectives of the article
Response: The conclusion has been revised, clarified and brought into full compliance with both the updated title of the article and the key results of the analysis. It now reflects the practical significance of the concept of metabolic instability and its place in the pathogenesis of endometrial cancer.
11. Comment: Authors stated that no funds required but in the authors contribution it mentioned that “A.F.; funding acquisition” Hope all the authors have contributed equally to this review article
Response: Thank you for your careful reading. The note about "funding acquisition" was indeed entered by mistake and does not correspond to the actual situation. No financing was involved. Correction has been made.
Thank you for noting that the visual materials presented in the original version did not fully correspond to the conceptual content of the article. In the updated version of the manuscript, a new scheme has been added that demonstrates the pathological cycle of metabolic instability and its alleged role in the formation of precancerous conditions of the endometrium. We are confident that the added illustration enhances the visual part of the work and contributes to a better understanding of the pathophysiological process under discussion.
We sincerely thank you for your in-depth and constructive analysis of our review. Your comments have played a key role in increasing its scientific rigor, structural coherence, and conceptual accuracy. We are grateful to take into account any subsequent recommendations, if they arise in the course of further work on the manuscript.
Kind regards,
Maria Sukhanova
(on behalf of all co-authors)